# Pharmacogenetics and Schizophrenia—Can Genomics Improve the Treatment with Second-Generation Antipsychotics?

**DOI:** 10.3390/biomedicines10123165

**Published:** 2022-12-07

**Authors:** Olga Płaza, Piotr Gałecki, Agata Orzechowska, Małgorzata Gałecka, Justyna Sobolewska-Nowak, Agata Szulc

**Affiliations:** 1Department of Psychiatry, Faculty of Health Sciences, Medical University of Warsaw, Partyzantów 2/4, 05-800 Pruszków, Poland; 2Department of Adult Psychiatry, Medical University of Łódź, Aleksandrowska 159, 91-229 Łódź, Poland; 3Department of Psychotherapy, Medical University of Łódź, Aleksandrowska 159, 91-229 Łódź, Poland

**Keywords:** pharmacogenetics, schizophrenia, atypical antipsychotics, second-generation antipsychotics, SNP, aripiprazole, risperidone, olanzapine

## Abstract

Schizophrenia (SCZ) is a complex psychiatric disorder of multifactorial origin, in which both genetic and environmental factors have an impact on its onset, course, and outcome. Large variability in response and tolerability of medication among individuals makes it difficult to predict the efficacy of a chosen therapeutic method and create universal and precise guidelines for treatment. Pharmacogenetic research allows for the identification of genetic polymorphisms associated with response to a chosen antipsychotic, thus allowing for a more effective and personal approach to treatment. This review focuses on three frequently prescribed second-generation antipsychotics (SGAs), risperidone, olanzapine, and aripiprazole, and aims to analyze the current state and future perspectives in research dedicated to identifying genetic factors associated with antipsychotic response. Multiple alleles of genes involved in pharmacokinetics (particularly isoenzymes of cytochrome P450), as well as variants of genes involved in dopamine, serotonin, and glutamate neurotransmission, have already been identified as ones of significant impact on antipsychotic response. It must, however, be noted that although currently obtained results are promising, trials with bigger study groups and unified protocols are crucial for standardizing methods and determining objective antipsychotic response status.

## 1. Introduction

### 1.1. Genetics and Schizophrenia

Schizophrenia (SCZ) is a neurodevelopmental disorder associated with deficits in cognition, affect, and social functioning, with prevalence in the general population, estimated to be at 0.5–1% [1]. The incidence is significantly greater in relatives of those diagnosed with schizophrenia—in children with either one or both of the parents diagnosed, the risk is at 17% and 40% respectively [2,3] whereas the rate of concordance in twins reaches 7% in case of fraternal siblings and is estimated at 41–65% in monozygotic ones [4]. Research points to the general heritability of schizophrenia to be at 79–85% [4,5].

Approaches in the search for genetic causes of schizophrenia changed in parallel to the development of new methods of analyzing the genome. Linkage analysis—the first DNA-based method—searched for chromosomal segments related to the disease. The base assumption of such analyses is that genetic markers located nearby on the same chromosome are unlikely to be separated into different chromatids during meiosis, therefore analyzing samples provided by members of a family with a history with SCZ and later creating a pedigree chart based on obtained results can help identify phenotypes more frequent in patients with SCZ and of a known genomic location, thus suggesting that genes related to SCZ are located nearby on the genome. Although linkage analysis allowed for the identification of chromosomal regions related to an increased chance of schizophrenia development [6], given the small sample sizes and statistical limitations, results obtained from linkage studies do not allow for a unanimous definition of risk loci and studies of genetic linkage should be used in combination with other methods of research [7].

The “candidate gene” method focused on analyzing whether alleles of pre-selected genes are involved in the pathogenesis of SCZ. Investigations focused on genotyping markers in genes selected either due to their position (identified for example in linkage studies) or functionality (for example genes encoding proteins involved in dopamine or serotonin transmission) and seeing whether certain alleles correlate with the disorder’s presence. Although studies allowed for the identification of multiple alleles, the variants were proven to have a statistically small effect on disease emergence, with general results considered to be somewhat disappointing and insufficient in explaining the genetic component in SCZ genesis, partially due to limitations of research methods [8].

What revolutionized research were genome-wide association studies (GWAS)—observational methods comparing the entirety of the genome of participants with varying phenotypes for a particular trait or disease. The base assumption of the method is that there is a correlation between the frequency of allele presence and the possibility of a genetic association—and as there are no pre-selected “genes of interest”, GWAS allows for the identification of SNPs (single nucleotide polymorphisms) throughout the entire genome, with recent technological advancements, such as microarrays and chips allowing for quick and inexpensive analysis of millions of SNPs. The most important study regarding the genetics of schizophrenia, carried out in 2014 by the Schizophrenia Working Group of the Psychiatric Genomics Consortium, analyzed all of the schizophrenia GWAS samples available at the time, which allowed for the identification of 128 independent associations in 108 loci of genome-wide significance, with 83 being new findings [9] What is particularly important, multiple associations were found in the gene for the dopamine 2 receptor, as well as genes crucial for synaptic plasticity, glutamatergic neurotransmission and genes related to central immune functions—results supporting theories connecting SCZ to dysfunctions in dopamine pathways [10] and abnormalities of the immune system [11].

However, given the noticeably lower than 100% concordance rate in monozygotic twins, researchers agree that although the cruciality of the genetic component in the etiology of SCZ is undeniable, other factors also have to play a role in SCZ development—a thesis supported by data proving the multifactorial origin of SCZ in which both genetic and environmental factors are assumed to be of importance [12,13]. This thesis stays in line with a theory that epigenetics—a field of study dedicated to the analysis of reversible changes in gene expression associated with DNA methylation, histone modifications, and regulatory means involving non-coding RNA, influenced, inter alia, by the organism’s exposure to external factors [14,15,16]—might be a potential bridge between genetics and environment and a crucial component in SCZ development [17,18].

### 1.2. Antipsychotics

Ever since their discovery, antipsychotics remain the key component of pharmacotherapy in SCZ, with their use being an inherent part of guidelines issued by the most esteemed medical groups, such as the National Institute of Health and Care Excellence [19] and the American Psychiatric Association [20].

Statistics regarding the efficacy of treatment remain inconsistent, with reports for remission rates for first-episode schizophrenia ranging from 17% to 78% and for multiple-episode patients from 16% to 62% [21], However, 34% of patients meet the criteria of treatment-resistant schizophrenia, meaning that symptoms persist despite 2 or more trials of antipsychotic medications of adequate dose and duration with documented adherence [22].

Second-generation antipsychotics are often used as a first-line treatment in schizophrenia—in comparison to typical, first-generation antipsychotics drugs (FGA), atypical, second-generation antipsychotics (SGA) have lower affinity and occupancy for the dopaminergic receptors, and high degree of occupancy of the serotoninergic and norepinephrine receptors—therefore, patients treated with SGAs on average experience less extrapyramidal effects. Nevertheless, SGAs may also cause serious adverse effects, including weight gain, metabolic syndrome, agranulocytosis or cardiological abnormalities, therefore they should be prescribed cautiously, as to ensure compliance and minimize patients’ discomfort.

In general, effective treatment of schizophrenia remains a challenge, as large variability in treatment response and tolerability among individuals makes it difficult to predict the outcome of a chosen treatment method and create more precise guidelines and algorithms for patients’ management—a unique situation in a disease of such prevalence and such big societal cost.

Given the developments in the field of genetics and molecular biology, pharmacogenetics research focused on investigating the possible impact of genetic variability on drug treatment efficacy, gained momentum in psychiatry, as it might make it possible to improve the results of treatment in patients with SCZ by providing physicians with additional means to create a more personalized treatment plan. Current research focuses on SNPs obtained throughout GWAS or candidate gene studies, with results later used to calculate polygenic risk scores (PRS)—probability estimates of an individual having a particular trait based on the presence of trait-associated alleles [23].

It has been proven that such approach can help predict how successful therapy with antipsychotics will be [24], however as of right now there are no panels or screening tests available to use in patients with SCZ that might allow detecting particular alleles with impact on response to antipsychotics.

As the amount of research in the field of psychiatric pharmacogenetics continuously grows and the methods allowing for genome analysis become both more available and relatively affordable, attempts to organize and systematize the existing knowledge and research results dedicated to schizophrenia seems justified. For this paper, two articles served as an inspiration—a 2021 review by Lisoway et al., focused on genetics and epigenetics in treatment with olanzapine, risperidone, clozapine and aripiprazole [17] and a 2014 review by Brennan—a very thorough analysis dedicated to “general” pharmacogenetics of second generation antipsychotics [25]. While the paper by Brennan served as a starting point of theoretical reference, work done by Lisoway and colleagues was a starting point for work dedicated to creating as up to date review as possible, with efforts to include papers either missed by or published after the publication of aformenetioned article.

This paper aims to review the current literature on the subject of pharmacogenetics of second-generation antipsychotics in patients with schizophrenia and discuss how the results obtained so far can be used in healthcare. Given the volume of research carried out on the matter, this review, although inspired by Lisoway et al., will only focus on olanzapine, risperidone and aripiprazole, as they can be considered representative examples of atypical antipsychotics. As the goal of this paper is to serve as a potential point of reference when considering potential legitimacy of genotype mapping and to present potential perspectives in widespread genetic testing of patients with schizophrenia, discussed polymorphisms will be mainly of genes related either to main metabolic pathways of the three drugs or serotonin and dopamine signaling, as those are the genes most universally analyzed in regard to pharmacogenetics and therefore most likely to be included in potential guidelines.

## 2. Materials and Methods

Four electronic databases (PubMed, Embase, Cochrane Library and Web of Science) were searched for papers in a period of August and September of 2022, using the keywords: “schizophrenia”, “polymorphism”, “pharmacogenetics” OR “pharmacokinetics” OR “pharmacodynamics”, “olanzapine” OR “risperidone” OR “aripiprazole” OR “second generation antipsychotics” OR “atypical antipsychotics”.

Of all the obtained results, a study was eligible for inclusion if the following criteria were met: an observational study design (clinical trial or randomized controlled trial) or a systematic analysis (meta-analysis or systematic review), work published either in English or in Polish, the focus of the paper dedicated directly to pharmacogenetics of risperidone, olanzapine and aripiprazole or more generally to pharmacogenetics of atypical antipsychotics with focus on risperidone, olanzapine or aripiprazole and the work itself published in a peer-reviewed journal.

Duplicates were excluded, as were articles which didn’t meet inclusion criteria or proved substantively insufficient throughout the full-text screening.

Additionally, reference lists of all relevant original and review articles were searched manually in order to identify additional eligible studies.

## 3. Results

### 3.1. Risperidone

#### 3.1.1. Genes Related to Pharmacokinetics

The most studied gene regarding pharmacokinetics concerning the effectiveness of treatment with risperidone is CYP2D6—a member of the cytochrome P450 mixed-function oxidase system, and the main enzyme responsible for risperidone metabolism [26]. CYP2D6 catalyzes the hydroxylation of risperidone to 9-hydroxyrisperidone—an active metabolite, also used as an antipsychotic drug under the name paliperidone, with approximately the same receptor binding affinity as risperidone [27]. Of all CYPs, CYP2D6 shows the biggest phenotypical variability, which impacts it’s effectiveness. Depending on the alleles of the gene present, patients can be separated into 4 groups—poor metabolizers (PM; little or no CYP2D6 function), intermediate metabolizers (IM; metabolize drugs at a rate somewhere between the poor and extensive metabolizer), extensive metabolizers (EM; normal CYP2D6 function) and ultrarapid metabolizers (UM; multiple copies of the CYP2D6 gene are expressed, so greater-than-normal CYP2D6 function occurs). Alleles determining the functionality of CYP2D6 can be found in Table 1. In PM patients both higher risperidone blood levels and higher Risperidone/9-hydroxyrisperidone ratio were observed [28] with similar results obtained in a study dedicated to patients identified as IM, where a higher plasma risperidone/9-hydroxyrisperidone ratio was observed [29]. Additionally, it has been proven that healthy volunteers identified as IMs or PMs were at higher risk of developing adverse effects when treated with risperidone [30] with extrapyramidal effects more probable in IM patients [31]. Kang et al., however, observed no correlation between CYP2D6 polymorphisms and risperidone pharmacokinetics [32].

Considering the consistency of the results obtained in various studies dedicated to the subject, guidelines recommending CYP2D6 genotyping for patients undergoing risperidone treatment were published by both Clinical Pharmacogenetics Implementation Consortium and the Dutch Pharmacogenetics Working Group [36], with DPWG recommending a reduction of risperidone dose in CYP2D6 PMs and changing treatment to an alternative drug or titration of the dose according to the maximum concentration for the active metabolite for CYP2D6 UMs.

Although other genes related to cytochrome P450 function—particularly those responsible for CYP3A5 isoenzyme, involved in risperidone metabolism as well—were also analyzed in regard to a potential relationship between phenotype and its impact on the plasma level of risperidone and its metabolites. Obtained results were inconsistent—Kang et al. found that homozygotes for the *3 allele, have a significantly decreased activity of the enzyme, which results in higher risperidone blood ratio [32], similarly copresence of such homozygotes with carriers of the CYP2D6*10 allele significantly increases risperidone blood level [37]. At the same time Vanderberghe et al. observed no statistically significant impact of CYP3A5 alleles on risperidone metabolism [38].

ABCB1 is a gene coding P-glycoprotein 1—a membrane transporter allowing for the transport of molecules across the cell membrane against the concentration gradient. Given its crucial role in metabolism, the gene itself and its polymorphisms have been a subject in candidate gene studies. In two polymorphisms of the gene, C3435T and G2677T/A (SNPs rs1045642 and rs2032582 respectively) heterozygosity was associated with bigger susceptibility to extrapyramidal symptoms when compared to homozygotic patients treated with risperidone [39,40]. Additionally, alleles of aforementioned polymorphisms have been associated with a lower risk of weight gain, with researchers claiming C3435 and G2677 are associated with higher activity of ABCB1 and in result lower weight gain but at the same time lower concentration of risperidone in the brain [40].

However, as if to further emphasize the need for more research dedicated to the subject, there are also studies which show that polymorphisms of ABCB1 have no or very limited impact on risperidone metabolism [41]. Therefore, in order to establish whether screening patients for ABCB1 polymorphisms is legitimate, additional research is necessary.

Haplotypes of COMT gene—coding catechol-O-methyl transferase, a protein involved in dopamine metabolism—were also proven to impact patients’ response to risperidone. One of the most studied polymorphisms—related to a SNP resulting in a valine to methionine substitution—impacts COMT effectiveness. Depending on polymorphism present, COMT activity changes—Val/Val genotype carriers present with high activity, heterozygotes show intermediate activity and Met/Met homozygotes present the lowest [42]. Presence of the SNP and resulting presence of methionine has been predictive of more significant improvement, particularly in regard to negative symptoms [43] in patients treated with risperidone. Cabaleiro et al. observed that, in healthy volunteers, carriers of Met allele had lower 9-hydroxyrisperidone AUC in comparison to Val/Val homozygotes [28], which they claim may indicate lower dopaminergic antagonism and consequent bigger improvement in negative symptoms in such patients when treated with risperidone. Another functional polymorphisms of the gene was associated with improved risperidone response, particularly in males [44,45].

#### 3.1.2. Genes Related to Pharmacodynamics

In serotonin receptors, haplotypes of HTRT2A were shown to impact the response in patients treated with risperidone. C/C homozygotes of the HTR2A variant (rs6313) showed better response to treatment in regard to negative symptoms [46] in comparison to carriers of at least one T allele. The results were however contradicted by Alladi et al. whose research showed there is no difference in risperidone response depending on the allele present [47]. When observing potential impact of HTRT2A type on response time in monotheraphy with risperidone or olanzapine, Maffioletti et al. observed that presence of T allele can be prognostic of better response, however it must be noted that effects were only statistically signifant when correlated with olanzapine response, and risperidone-specific analysis show no statistically significant impact [48].

Alleles of another serotonin receptor—HTR6—were also associated with risperidone response. In a group of 201 Chinese patients with schizophrenia, a statistically important difference in reaction to risperidone treatment was observed between variants of the gene, with presence of A allele associated with improved therapeutic response and A/A homozygotes showing considerably better improvement in positive symptoms [49]. However, results acquired by Ikeda et al. after 120 first-episode neuroleptic-naive schizophrenia patients were treated with risperidone monotherapy, showed that among all serotonin receptors there are no variants associated with the efficacy of risperidone treatment [50].

Similar results were obtained in candidate gene studies dedicated to HTR2C—in studies dedicated to risperidone related weight gain, no polymorphism of impact were distinguished [40].

Although they found that variants of genes encoding serotonin receptors made no difference in risperidone treatment, in their research Ikeda et al. proved that there are variants of DRD2 genes associated with a better response to risperidone. Two SNPs were discovered as variants of significant impact—in both cases homozygotes (A/A in rs1799978 and A1/A1 in rs1800497) were showing significantly better improving in PANSS scores [50]. The A1/A1 homozygotes also exhibited higher prolactin concentrations and hyperprolactinemia associated adverse events [25,51] Another SNP analysis showed that in C/C homozygotes, better improvement in positive symptoms was observed in comparison to carriers of the T allele [52], additionally such male homozygotes experienced lower elevation of prolactin levels during treatment—a result consistent with findings from earlier research [25,51].

Interestingly, the same C allele is both relatively common in general population and has been found by the Psychiatric Genomics Consortium genome-wide association study to be a risk allele for schizophrenia—relatively worse response to treatment in carriers of the “safe” T allele researchers explain with it’s possible modifying role on dopamine signaling [52].

Summary of gene polymorphisms with potential impact on risperidone pharmacokinetics and pharmacodynamics can be found in Table 2.

### 3.2. Olanzapine

#### 3.2.1. Genes Related to Pharmacokinetics

The main enzyme involved in the oxidative pathway in the breakdown of olanzapine is CYP1A2. Candidate gene studies allowed for 2 alleles which impact metabolism and therefore plasma concentration of olanzapine to be defined—according to Czerwensky et al. carriers of the *1D and *1F polymorphisms (rs35694136 and rs762551 respectively) present significantly different serum concentration and dose-corrected serum concentration of olanzapine depending on precise genotype. CYP1A2*1D related delT allele was associated with an increase of serum concentration of olanzapine, with particularly significant increase in dose-corrected serum concentrations and dose- and body weight-corrected serum in homozygotes, whereas CYP1A2*1F AC heterozygotes exhibited highest serum- and dose-corrected serum concentrations of olanzapine [54]. However, other studies focused on candidate genes found no such correlation [55,56].

Similarly, in works dedicated to analyzing polymorphisms of CYP3A5 results presented on the matter were contradictory, with Cabaleiro et al. pointing out a significantly smaller AUC in *3/*3 homozygotes in comparison to carriers of at least one *1 allele [55], whereas other studies show there is no significant impact on olanzapine pharmacokinetics between variants of CYP3A5 [57]. Both teams however reached a consensus when researching CYP2C9, pointing out that patients defined as PMs (CYP2C9*2 and CYP2C9*3) in regards to this isoenzyme presented with a lower rate of metabolism of olanzapine (increased half-life and bigger volume and distribution) and consequent higher incidence of accumulation-related adverse effects—in particular dizziness [55,56]. In research dedicated to CYP2D6, there was no statistically significant difference in the plasma level of olanzapine and its metabolites observed between phenotypes [56,58]. Given the lack of concordance in the results of studies on the potential impact of polymorphisms of cytochrome P450, as of right now it is impossible to create guidelines or recommendations regarding the legitimacy of profiling patients at this angle, similar to recommendations of DPWG and CPIC regarding risperidone therapy, and further research is needed.

Besides cytochrome P450, flavin-containing monooxygenases are the most important oxidative enzymes involved in olanzapine metabolism. Söderberg et al. investigated the potential role of polymorphisms in genes encoding FMO3—the main isoenzyme expressed in the liver of adult humans—and FMO1 in the metabolism of olanzapine. A FMO1*6 polymorphism, with a presence of at least one A allele was associated with increased dose-adjusted serum olanzapine concentration, whereas G/G homozygotes of one of FMO3 polymorphisms present with up to 50% lower dose-adjusted serum olanzapine N-oxide concentrations [57].

Haplotypes of UGT1A4—gene encoding UDP-glucuronosyltransferase, an enzyme of the glucuronidation pathway—were also the subject of various studies, given the important role of the protein in medication metabolism and considerable interpatient variability in its hepatic clearance. It has been observed that concentration of olanzapine was lower and concentrations of olanzapine 10-N-glucuronide—the main metabolite of olanzapine—were significantly higher in heterozygous and homozygous carriers of the G allele [59,60], same allele was associated with an increased sympathetic nervous activity in its carriers, and consequently lower incidence of sympathetic dysfunction. [61]. Of other genes involved in the glucuronidation pathway, an *2 allele of UGT2B10 was associated with a decrease in vitro glucuronidation [60], whereas alleles of UGT1A1 influenced Tmax—with highest observed in T/T homozygotes, [56] and C/C homozygotes reported to be at higher risk of elevated glucose blood levels [62].

Out of many polymorphisms of the ABCB1 gene, three were identified as having an impact on olanzapine pharmacokinetics. Skogh et al. observed that 3 SNPs (C1236T, rs1128503; C3435T, rs1045642; G2677A/T, rs2032582) impact serum and CSF olanzapine concentration, with significantly higher levels observed in triple T homozygotes compared to other haplotypes [63]. The C3435T polymorphism was also associated with lower clearance and volume of distribution in T/T homozygotes, although statistical importance was achieved after correction for CYP3A5*3 influence [53]. Same variant of the polymorphism was associated by Lin et al. with positive percent change in Brief Psychiatric Rating Scale score, as well as negative symptom reduction and olanzapine plasma concentration, although only the change in BPRS was statistically significant [64].

Zubiaur et al. observed higher exposure and reduced clearance in C/C homozygotes of C1236T, higher exposure was also associated with C/C homozygotes T193C. This haplotype was also associated with asthenia and palpitations, although it must be noted that while statistically significant, both adverse effect were recorded in only 1 person each (80 participants in total) [57]. Authors however emphasise that given contradictory results of previously carried out research, including aforementioned paper by Sainz-Rodriguez et al. [53], additional studies on the matter are necessary.

#### 3.2.2. Genes Related to Pharmacodynamics

Multiple different research papers describe a relationship between variants of the HTRC2 gene and response to olanzapine. Multiple genes with associated with statistically significant impact on weight gain (A1165G, rs498207; G997A, rs3813928; C759T, rs3813929) and metabolic syndrome (rs1414334; G697C, rs518147) [65,66,67]. Multiple researchers point out a trend between C alleles and significant positive association to weight gain, particularly in case of C759T polymorphism, although additional data is necessary on the matter.

When researching HTR2A, the rs6314 polymorphism has been a subject of multiple analyses carried out by Blasi et al., who chose this particular one due to its impact on hippocampal volume and activity, episodic memory performance and diagnosis to schizophrenia. Presence of T allele of the gene was associated with decreased prefrontal messenger RNA expression in postmortem prefrontal cortex, inefficient prefrontal blood oxygen level impaired working memory and attention behavior, as well as with reduced improvement in negative symptoms after olanzapine treatment [68]. Follow up research proved negative impact of T allele presence on both treatment efficacy and clinical presentation, it must however be noted that observed results reach statistical significant in analyzed variants when concomitant with particular variants of the DRD2 gene (rs1076560 C/C homozygotes) [69] Mentioned before the study by Maffioletti et al. also pointed out the impact that variants of the gene may have on the efficacy of treatment with olanzapine in combination with risperidone [35], however, similar to risperidone, the obtained results were statistically insignificant once olanzapine-specific calculations were performed.

Given the biggest affinity of olanzapine to DRD2 out of all dopamine receptors, the polymorphisms and variants of the gene are continuously studied to establish the potential impact it may have on the efficacy of therapy. Aforementioned study by Blasi et al. identified that lack of T alleles of the rs1076560 variant may be prognostic of better effectiveness if coexistent with rs6314 C/C genotype [69]. In a clinical trial with patients experiencing their first episode of schizophrenia, it was observed that two polymorphisms in the promoter region of the gene may be used as predictors of response time—carriers of G allele in the A241G (rs1799978) presented with significantly faster response time to treatment [70]. What is interesting, as mentioned in the risperidone dedicated subchapter of this paper, the tendency was reversed when analyzing factors impacting efficacy of treatment—there A/A homozygotes presented with better response to treatment [71]. The second SNP,—141C Ins/Del (rs1799732), was prognostic of a delayed response time when the deletion was present [70]. Same allele was associated with increased chance of somnolence [57]. Multiple alleles predisposing carriers to increased prolactin levels, with studies dedicated to rs2734842, rs6275 and rs6279 in females presenting with consistent results and defining C, T and C alleles respectively as alleles of increased risk [72]. Alleles of the Taq1A gene remain alleles of interest, although further research is needed to achieve statistically significant results [73].

Several DRD3 alleles were also found to be of probable importance in the tolerance and effectiveness of olanzapine. In the trial run by Koller et al. increased level of prolactin was associated with Gly/Gly homozygotes of the Ser9Gly (rs6280) variant of the receptor [62] and the same polymorphism was associated with a significant improvement in positive symptoms in 2 independent studies [74,75].

Summary of gene polymorphisms with potential impact on lanzapine pharmacokinetics and pharmacodynamics can be found in Table 3.

### 3.3. Aripiprazole

#### 3.3.1. Genes Related to Pharmacokinetics

Metabolism of aripiprazole is dependent mainly on CP2D6 and CYP3A4, therefore polymorphisms of those two enzymes remain the main point of interest in research (Urichuk et al., 2008).

Polymorphisms of CYP2D6 were studied extensively, with obtained results mainly consistent, particularly regarding polymorphisms’ impact on plasma levels of aripiprazole. In a meta-analysis of 10 studies published through February 2018, Zhang et al. observed a statistically significant difference in aripiprazole plasma concentration between EMs and IMs, but no such dependence was recorded between IMs and PMs. Interestingly, no statistically significant difference was observed between aripiprazole plus dehydroaripiprazole serum levels among EMs, IMs and PMs. Somewhat logically, inversely proportional relationship between the total activity of the alleles present and general sum of aripiprazole and dehydroaripiprazole serum levels, although highest serum levels of solely dehydroaripiprazole were observed in EMs [35] Similarly, a meta-analysis of Milosavljević et al., based on the data derived from 12 unique studies, including 1038 individuals, with particularly important differences in serum levels of aripiprazole between patients with either PMs or IMs genotype versus EMs [78]. According to the authors, obtained results were so statistically significant and precise, they can be used in creating general guidelines for aripiprazole treatment and the potential usefulness of genotype analysis as part of it, as they prove the need for dose reduction in patients with genetically proven decreased CYP2D6 function. As of today, such guidelines were published by FDA and the DPWG, however, they only concern patients with PMs haplotypes, with both bodies recommending dose reduction to avoid side effects [79].

While results obtained from CYP2D6-focused research were mostly consistent, comparing the results of studies on CYP3A4 variants in aripiprazole metabolism and treatment tolerance proved more challenging. Given the inducibility of the enzyme, researchers found it more difficult to get statistically significant and unambiguous results, although FDA recommendations suggest decreasing the dose of aripiprazole if patients are prescribed CYP3A4 inhibitors at the same time. Obtaining concise and undisputable results was also impacted by similarities between CYP3A4 and CYP3A5—similarly, results of studies dedicated to CYP3A5 were also inconsistent. Although Belmonte et al. observed a correlation between polymorphisms and the increased frequency of adverse drug reactions (nausea) in *1/*1 homozygotes, with carriers of *1 allele also presenting with higher ratio dehydro-aripiprazole/aripiprazole ratio in comparison to *3/*3 homozygotes [80], Suzuki et al. found no polymorphisms with an impact on treatment tolerance or metabolism of aripiprazole [81]. However, it must be noted that test groups were vastly different—Belmonte research included healthy Spanish volunteers, whereas Suzuki analyzed the genomes of Japanese patients with schizophrenia, thus comparing results is near impossible.

Despite less significant role of ABCB1 in metabolism of aripiprazole, given the nevertheless crucial role in general metabolism of xenobiotics and promising results of analyses dedicated to other SGAs, potential impact of ABCB1 polymorphisms on aripiprazole treatment is also studied. One study, including 90 pediatric patients treated with aripiprazole, focused on 2 SNPs of known impact on SGAs metabolism—C3435T and G2677T/A (SNPs rs1045642 and rs2032582 respectively). Results show that when a patient has a TT/TT genotype, statistically significant reduction of aripiprazole Ct/ds is observed. There is no impact of phenotype of those SNPs on adverse drug reactions [82]. However, majority of research dedicated to the matter was unable to identify polymorphisms of statistically significant impact on either pharmacokinetics or tolerance of aripiprazole [53,80,81] or as in the case of Koller et al. despite variants of which homozygotic carriers had significantly higher prolactin concentrations (rs10280101 A/A, rs12720067 C/C and rs11983225 T/T), association was absent when haplotypes were analyzed.

#### 3.3.2. Genes Related to Pharmacodynamics

Located nearby the DRD2 gene, and often considered a part of it when analyzed in its context, Taq1A polymorphism has been linked to reduced D2 receptor density in the caudate and putamen, therefore potential impact of polymorphisms in this area of the genome is of special interest to scientists researching pharmacogenomics. Carriers of the A1 allele showed a tendency of responding better to aripiprazole in comparison to noncarriers in a study of thirty Japanese patients with schizophrenia, although the observed differences did not reach a statistically important level. In the same study, a statistically important difference was observed in a decrease in plasma levels of homovanillic acid—a marker of metabolic stress in schizophrenia—with changes of pHVA differing between responders and nonresponders in the A1 allele carriers (but not in the A1 allele noncarriers) However, the authors note that given the small sample size, results are limited and should be verified in a bigger group of patients [83]. Kwon et al. also reported better response to aripiprazole in carriers of the A1 allele, with significant differences observed in PANSS score following the course of the study in A1 carriers, however, there were no significant differences in the changes in the CGI, AIMS, and BAS scores [84]. Bigger PANSS score improvement following aripiprazole treatment in A1 carriers was also reported by [85]. The same study allowed for identification of another polymorphism—C957T (rs6277), which when present in a C/C homozygotic form was predictive of poorer improvement in PANSS excitement score when compared with T allele carriers.

In a study dedicated to HTR2A polymorphisms and their potential impact on aripiprazole treatment efficacy, Chen et al. reported 2 polymorphisms of impact—1438G (rs63311) and T102C (rs6313), with GG/CC genotype predictive of poorer negative symptoms improvement [86].

Amount of research dedicated to HTR2C polymorphisms and efficacy of aripiprazole treatment is as of today limited, with analyses carried out so far finding no significant correlation between C759T (rs3813929) variants on aripiprazole kinetics [56] or increased risk of metabolic syndrome [62].

Summary of gene polymorphisms with potential impact on aripiprazole pharmacokinetics and pharmacodynamics can be found in Table 4.

### 3.4. Summary of Results

Although considerable work has been devoted to identification of reliable biomarkers to predict antipsychotic response in SCZ, data obtained is still insufficient and with the exception of singular instances, there are no general guidelines regarding potential application of pharmacogenetics in daily clinical practice.

The exceptions concern CYPD26 and CYP3A4 enzymes. In treatment with risperidone, Dutch Pharmacogenetics Working Group recommends a reduction of dose in patients with the PMs phenotype of CYP2D6 and suggests either a change of treatment or titration of the dose in those who can be classified as CYP2D6 UMs. The same consortium, together with the FDA, recommends that patients who can be classified as PMs in regard to CYP2D6 and are treated with aripiprazole, should be administered a reduced dose. Additionally, FDA suggests a reduction of aripiprazole dose when taken simultaneously with CYP3A4 inhibitors, with no specification of patients’ phenotype.

Nevertheless, although undeniably useful and important, such recommendations, are in no way sufficient, thus further emphasizing the need for continuous research in the matter.

As this review shows, there are multiple genes that continue to be a subject of analyzes in search for biomarkers of antipsychotic response. Enzymes of the cytochrome P450 group are particularly of interest given a large number of polymorphisms and isoforms of various functions and their crucial role in metabolism of antipsychotics. The discovery of polymorphisms conditioning the activity of enzymes made it possible to create a new classification, used inter alia in creation of aforementioned guidelines, with a potential of being utilized again on basis of results from other research, including for example work by Czerwensky et al. or Cabaleiro et al., who studied how olanzapine blood level can be impacted by CYP1A2 and CYP3A5 polymorphisms respectively.

Another enzyme crucial in metabolism of antipsychotics and therefore crucial in efficacy of treatment is ABCB1, with this review covering studies dedicated to multiple polymorphisms and potential impact on treatment with one of 3 analyzed antipsychotics. Researchers managed to identify multiple alleles with a statistically significant impact on pharmacokinetics of the treatment, as well multiple variants directly correlated with an increased risk of adverse effects. However, as is the general trend in studies dedicated to pharmacogenetics so far, papers on the matter present often mutually exclusive results, which is linked to small study groups, different research protocols and lack of unambiguous standards.

The obtained results also show that the role of enzymes involved in the glucuronidation pathway is undeniable in metabolism of antipsychotic drugs, with various polymorphisms being predictors of treatment efficacy, particularly in therapy with risperidone.

Of all the genes encoding dopamine receptors, most work so far has been dedicated to DRD2, which allowed for identification of both “predictor alleles”—prognostic of good response to treatment—as well as “risk alleles”—related to an increased risk of adverse drug reactions—in research dedicated to all 3 of preselected atypical antipsychotics. Results regarding DRD3 polymorphisms remain mostly inconsistent, with only singular alleles identified and their impact deemed statistically significant.

Given the mechanism of action of atypical antipsychotics, serotonin receptors and their polymorphisms have been a subject of multiple trials and analyses, with the biggest number of findings published being dedicated to HTR2C and HTR2A polymorphisms. However, as of today there are no conclusions strong enough to justify issuing recommendations based on them. Nevertheless, results obtained so far are promising, therefore research focused on those receptors in particular should continue.

## 4. Discussion

One of the top 15 leading causes of disability worldwide, schizophrenia is a debilitating mental illness associated with huge health, social, and economic concerns. Life expectancy of patients diagnosed with schizophrenia is approximately 25 years shorter in comparison to healthy control with co-occurring medical conditions, such as heart disease, liver disease, and diabetes more prevalent in this group of patients, and this group of patients significantly more likely to commit suicide [87]. Costs associated with schizophrenia are high both on the personal and societal level—despite somewhat low prevalence, schizophrenia is considered to be an illness of a disproportionately high societal impact, linked both to cost related directly to financing treatment and general health care, as well as hidden costs, associated with lost productivity, social service needs, criminal justice involvement, and other factors unrelated to health care [88].

The exact pathophysiology of schizophrenia is still not fully understood, which directly correlates with difficulties in creating universal guidelines for pharmacotherapy. As of today, many antipsychotic medications are still prescribed via the “trial-and-error” method [17], a manner potentially exposing the patient to severe side effects, and healthcare to unnecessary costs should they present. In search for ways to optimize treatment with antipsychotics, recent advancements in the field of pharmacogenetics seem to hold a lot of promise.

The papers analyzed in this review allow to state unequivocally that there are multiple alleles of genes directly related to SGAs response treatment—both those linked to pharmacokinetics, as well as genes associated with pharmacodynamics such as those involved in dopaminergic and serotonergic neurotransmitter systems. What is more, although they have not been the focus of this paper, multiple genes unrelated to traditional theories of schizophrenia genesis have also shown significant correlation with antipsychotic response [89].

However, even though the results obtained so far seem promising, scientists agree that there are multiple limitations in the field of pharmacogenetics that should be addressed in order to further progress the research.

First and foremost, the candidate gene approach, being hypothesis driven, significantly restricts possible results, making it impossible to identify novel loci, previously not associated with the disease, that might nevertheless somehow impact the response efficacy through polygenic measures. Therefore GWAS, with consequent calculations of PRS, should be made the leading method in pharmacogenetic research in upcoming years, particularly given the success rates of research carried out this way so far [24].

Additionally, the aim should be to conduct big, multi-centered trials, with clear protocols of conduct, comprehensive patient assessments, using of consistent scales, and questionnaires focusing on objective measures. This will not only allow for researchers to standardize methods to determine antipsychotic response status—a crucial prerequisite for creating guidelines of conduct, but also it will make it possible for other research teams to copy the earlier used methods when trying to replicate results—a mission significantly more difficult to achieve in case of research carried out on small, homogenous groups of patients, each run on different, inconsistent research protocols. A prime example of such challenge would be comparing mentioned before trials carried out by Suzuki et al. and Belmonte et al.—although focused on the same polymorphisms, the analyzed groups and study protocols were so vastly different that even standardizing the results seems nearly impossible.

As of today, there are no general guidelines regarding pharmacogenetic testing in antipsychotic treatment in schizophrenia, with tools allowing for such testing unavailable commercially. In major depressive disorder, however, pharmacogenetic tests are proving to be useful tools, further providing support for ongoing effort to introduce common pharmacogenetic testing in acute treatment in major depressive episode [90]. Facts that such guidelines are created for treatment with antidepressants, with the work significantly more advanced mostly due to the bigger amount of data dedicated to the subject, further emphasizes the need for large randomized controlled trials dedicated to the role of gene polymorphisms in antipsychotic treatment efficacy, as their results may truly revolutionize psychiatry and treatment of schizophrenia.

## Figures and Tables

**Table 1 biomedicines-10-03165-t001:** Alleles of CYP2D6 gene and their impact of functionality: [33,34,35].

**Type of Metaboliser**	**Level of Function and Genotype**
PM (poor)	no function—individual carrying only no functional alleles
IM (intermediate)	decreased function—individual carrying one allele of decreased function and one non-functional allele
EM (extensive)	normal function—individual carrying two alleles of decreased function or one allele of normal function and one non-functional allele or one allele of normal function and one allele of decreased function
UM (ultra-rapid)	increased function—individual carrying duplicates of functional alleles
**Level of function**	**Example of allele**
Normal function	*1 (wild type), *2 (rs16947 or rs1135840), *35, *39 (rs1135840), *43, *45
Decreased function	*9 (rs5030656), *10 (rs1065852), *17 (rs28371706 or rs16947), *29 (rs61736512 or rs1058164 or rs16947 or rs59421388 or rs1135840), *41 (rs28371725)
Non-functional	*3 (rs35742686), *4 (rs3892097), *5 (whole gene deletion), *6 (rs5030655), *7 (rs5030867), *8 (rs5030865), *11 (rs5030863), *12 (rs5030862), *20 (rs72549354) and others

**Table 2 biomedicines-10-03165-t002:** Summary of gene polymorphisms which might impact risperidone treatment.

Gene	SNP (and Precise Genotype if Applicable)	Association	Observed by	Not Observed by
**Genes related to pharmacokinetics of risperidone**				
CYP2D6	PM	higher risperidone serum levels higher risperidone/9-hydroxyrisperidone serum ratio	[28]	
		higher risk of developing adverse effect	[30]	
	IM	higher risperidone/9- hydroxyrisperidone serum ratio	[29]	
		higher risk of developing adverse effects	[30,31]	
CYP3A5	rs776746 (*3/*3 homozygotes)	higher risperidone serum levels higher risperidone/9-hydroxyrisperidone serum ratio	[32,37]	[38]
ABCB1 (MDRD1)	rs2032582 (GT and GA heterozygotes) rs1045642 (CT heterozygotes)	higher risk of akathisia and dystonia, bigger weight gain	[39,40]	[41]
	rs2032582 allele (G allele carrier) rs1045642 (C allele carrier)	enhanced risperidone elimination, smaller weight gain	[39,40,53]	[41]
COMT	rs9606186 (GG genotype)	improved response in male patients	[44,45]	
	rs4680 (Met allele carrier)	bigger improvement in negative symptoms, lower 9-hydroxyrisperidone AUC	[28,43]	
**Genes related to pharmacodynamics of risperidone**				
HTR2A	rs6313 (C/C homozygotes)	bigger improvement in negative symptoms	[46]	[47]
HTR6	rs6699866 (A/A homozygotes)	significant association with a reduction in positive PANSS scores	[49]	[50]
DRD2	rs1799978 (A/A homozygotes)	improvement in total PANSS score	[50]	[47]
	rs2514218 (C/C homozygotes)	bigger improvement in positive symptoms, lower elevation of prolactine in male patients	[52]	[47]
	rs1800497 (A1/A1 homozygotes)	elevated level of prolactine, improvement in total PANSS score	[25,50,51,52]	[47]

**Table 3 biomedicines-10-03165-t003:** Summary of gene polymorphisms which might impact olanzapine treatment.

Gene	SNP (and Precise Genotype if Applicable)	Association	Observed by	Not Observed by
**Genes related to pharmacokinetics of olanzapine**				
CYP1A2	rs35694136 (delT/ delT homozygotes)	dose-corrected serum concentrations and dose- and body weight-corrected serum	[54]	[55,56]
	rs762551 (AC heterozygotes)	increased serum- and dose-corected serum concentrations of olanzapine	[54]	[55,56]
CYP3A5	rs776746 (*3/*3 homozygotes)	lower AUC	[55]	[57]
UGT1A4	rs2011425 (G allele carriers)	lower olanzapine and higher olanzapine 10-n-glucuronide plasma levels	[60,76]	
		sympathetic nervous activity was significantly higher—decreased frequency of adverse side effects	[61]	
UGT2B10	rs61750900 (*2 allele carriers)	lower olanzapine 10-n-glucuronide plasma levels	[60]	
UGT1A1	rs887829 (T/T homozygotes)	higher Tmax	[56]	
	rs887829 (C/C homozygotes)	significantly higher glucose blood levels	[62]	
FMO3	rs887829 (C/C homozygotes)	decrease in dose-adjusted serum olanzapine N-oxide concentrations	[59], p. 1	
FMO1	rs12720462 (A allele carriers)	increased dose-adjusted serum olanzapine concentrations	[59], p. 1	
ABCB1	rs1128503, rs1045642, rs2032582 (T/T/T haplotype)	higher serum and CSF olanzapine concentration	[63]	
	rs1128503 (C/C haplotype)	higher exposure and reduced clearance of olanzapine	[57]	[53]
	rs1045642 (T/T homozygotes)	lower clearance and volume of distribution	[53]	
	rs3842 (C/C homozygotes)	higher olanzapine exposure increased risk of adverse effects—palpitations, asthenia	[57]	[53]
**Genes related to pharmacodynamics of olanzapine**				
HTR2C	rs498207 (AA/A genotype) rs3813928 (G allele carriers) rs3813929 (C allele carriers)	weight gain	[65,67,77]	
	rs1414334 (C allele carriers) rs518147 (C allele carriers)	increased risk of metabolic syndrome and weight gain	[65,66]	
HTR2A	rs6314 (CC homozygotes if rs1076560 CC present)	Better response to treatment (particularly in positive symptoms)	[68,69]	
DRD2	rs1799732 (Del carriers)	slower response to treatment	[70]	
		increased risk of adverse reactions	[57]	
	rs1076560 (CC homozygotes if rs6314 CC present)	Better response to treatment (particularly in positive symptoms)	[68,69]	
	rs2734842 (C allele carriers) rs6275 (T allele carriers) rs6279 (C allele carriers)	Elevated prolactine levels in female patients	[72]	
DRD3	rs6280 (Gly/Gly homozygotes)	better response to olanzapine (positive symptoms), higher blood levels of prolactin	[62,74]	

**Table 4 biomedicines-10-03165-t004:** Summary of gene polymorphisms which might impact aripiprazole treatment.

Gene	SNP (and Precise Genotype if Applicable)	Association	Observed by	Not Observed by
**Genes related to pharmacokinetics of aripiprazole**				
CYP2D6	UM, EM, IM, PM	Aripiprazole and its metabolites blood levels inversely proportional to the number of functional alleles	[28]	
		Significantly higher serum levels of olanzapine in IMs an PMs in comparison to EMs	[78]	
CYP3A5	wild type (*1 allele carriers)	higher dehydroaripiprazole/aripiprazole ratio	[80]	[81]
	Wild type (*1/*1 homozygotes)	Increased frequency of adverse reactions (nausea)	[80]	[81]
ABCB1	rs1045642 and rs2032582 (TT/TT homozygotes)	reduction of aripiprazole Ct/ds	[82]	[53,80,81]
**Genes related to pharmacodynamics of aripiprazole**				
DRD2	TaqIA (*A1 carriers)	better improvement in PANSS score	[73,84,85]	
	rs6277 (C/C homozygotes)	poorer improvement in PANSS excitement score	[85]	
HTR2A	rs6311 and rs6313 (GG/CC genotype)	predictor of poor response to treatment (particularly for negative symptoms)	[86]	

## Data Availability

Not applicable.

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
