# Peer review of "Pharmacogenetics and Schizophrenia—Can Genomics Improve the Treatment with Second-Generation Antipsychotics?"

_biomedicines, 2022, doi:10.3390/biomedicines10123165_

Round 1

Reviewer 1 Report

1. I was going to initially reject this paper without reading it – there is no Abstract, much of the paper is written in a bold hard to read font, then the font changes abruptly to something else, and again for the references. It is as if the authors just tried to get lots of references and review this area of the schizophrenia field, without making any effort with regards to the reader (i.e., me, the reviewer) who has to read it. I think it is very unfair for the authors to then say something  I imagine like “sorry…. we’ll reformat the article for you….” and then I have to spend another 20 hours reading it. (I normally spend about 20 hours on reviews, checking each line, checking references, etc.)

Again, articles without Abstracts and articles that the authors cannot even bother to put in a readable format should be rejected out of hand, with the professionalism of the authors questioned.

2. However, the article does touch upon an important subject – what antipsychotics do we give to the millions of patients around the world who need these medications?  This is an important area for research. A review article in this area is welcome, so I went ahead and read the article.

3. There was a review article that recently was written by researchers at CAMH/University of Toronto – Lisoway, Chen, et al 2021. In Lisoway’s article, they also consider the pharmacokinetics, pharmacodynamics of Risperidone, Olanzapine and Aripiprazole. While the authors do mention Lisoway 2021 at the end of the paper in the discussion, they really should mention Lisoway (and other equivalent reviews) at the beginning, and explain how their review is different. (For example, if they said their review was more up to date, that would be ok for me. However, to ignore other researchers who have done similar work, is a serious omission.)

4. Line 64 – The level of concordance in monozygotic twins is not that low, the authors actually mention how high it is on the previous page. However, I guess the authors want to say that it is not 100% and other factors need to be considered.

5. Lines 80-83 by themselves seem out of place. Seems like more material is needed here or else integrate them with the previous paragraph. Also, it should be noted, the Lisoway 2021 also considers pharmacoepigenetics.

6. Line 89 – No physician in Canada or USA prescribes antipsychotics because they are “recommended by the biggest medical associations” – that is not a valid reason, and would be considered political, and malpractice. Rather, antipsychotics are used as recommended by guidelines, as recommended by evidence-based medicine, etc that various organizations create. Some of these organizations are smaller than others—that does not matter, what matters is the level of evidence.

Author Response

  • Not sure if it is appropriate to start a response to a review with an apology, nevertheless, I am going to do it - I am really sorry for the form in which you received the paper. In my defence, the word file I uploaded is coherently and legibly formatted on my computer and it must’ve undergone some modifications during the transfer to the server - I was horrified when I re-downloaded the paper to edit it and saw the formatting and I assume that this was the way you first saw this article, therefore once again I want to apologise for not uploading it in the suggested format right away. The same goes for the abstract - it is my first time submitting work to this journal, I assumed that given the fact that the abstract is added separately, reviewers will also receive this. 
  • This version of the paper is edited using the template provided by the editors - I do hope that reading it will be a more visually pleasing experience
  • The impact of Lisoway’s paper on the creation of this article was explained in the introduction part, the paper itself was also cited more often
  • The line regarding the „low rate of concordance”  has been edited to simply reflect the importance of less than 100% concordance
  • The paragraphs about epigenetics have been significantly edited and serve more of a role of extra information for the reader
  • The line „recommended by the biggest medical association” is unfortunate and somewhat of a loan translation - it is modified to emphasise not the size but the professionalism of given groups

Reviewer 2 Report

Plaza et al. focused on 3 frequently prescribed second-generation antipsychotics - risperidone, olanzapine and aripiprazole - and aims to analyse the current state and future perspectives in research dedicated to identifying genetic factors associated with antipsychotic response. It is quite interesting study, but study is very disorganized. It requires severe reorganization. Please remove from the text all punctated parts with dots. The text is very heterogenous and tables are quite messy. Authors did not use uniform nomenclature for the variants description. It would be useful to discuss also other potential pharmacogenomic targets (Neurol Neurochir Pol 2022;56(1):4-13.). The reason why the authors choose only three drugs remains unclear.

Author Response

  • Not sure if it is appropriate to start a response to a review with an apology, nevertheless, I am going to do it - I am really sorry for the form in which you received the paper. In my defence, the word file I uploaded is coherently and legibly formatted on my computer. It must’ve undergone some modifications during the transfer to the server - I was horrified when I re-downloaded the paper to edit it and saw the formatting. I assume that this was the way you first saw this article, therefore once again I want to apologise for not uploading it in the suggested format right away. Same goes for the abstract - it is my first time submitting work to this journal, I assumed that given the fact that the abstract is added separately, reviewers will also receive this. 
  • The tables have been reorganised, with all following the same structure now
  • As uniform as possible nomenclature was used to describe variants with at least one way of defining the SNP used at all times
  • The reasons why those drugs and those particular genes are the subjects of this review are explained more thoroughly in the introduction part

Round 2

Reviewer 1 Report

1. Thank you for reformatting the paper and using the Journal template. It is readable now :)

2. Unfortunately, normally I would use the Journal editing system to see the changes made to the old text, but the entire original article which was bolded and in heterogeneous fonts is crossed out, so it is hard to see where changes were made. I mention this since it precludes me from reviewing articles as carefully as I normally would.

3. However, overall reading through the article now again, it is much more coherent, and thank you for talking about Brennan's work and Lisoway's work, and how this article improves upon them. Ok.... great. The topic of prescribing antipsychotics is an important one, and a more up to date review of the pharmacogenetics of three commonly (although many practitioners do not use these medications as often anymore, as newer medications have other advantages) prescribed medications is welcome. I believe as such at this point the article becomes more worthy of being published.

4. I am ok with the article being published at this point. As a review article, it is more coherent (although another edit pass by the authors should be considered), it cites previous review articles which were similar, it covers a relevant topic, and the ethical concerns I had previously have been addressed.